# Structure function and engineering of multifunctional non-heme iron dependent oxygenases in fungal meroterpenoid biosynthesis

Yu Nakashima[1], Takahiro Mori[1], Hitomi Nakamura[1], Takayoshi Awakawa[1], Shotaro Hoshino[1], Miki Senda[2], Toshiya Senda[2,3] & Ikuro Abe [1]

Non-heme iron and α-ketoglutarate (αKG) oxygenases catalyze remarkably diverse reactions using a single ferrous ion cofactor. A major challenge in studying this versatile family of enzymes is to understand their structure–function relationship. AusE from *Aspergillus nidulans* and PrhA from *Penicillium brasilianum* are two highly homologous Fe(II)/αKG oxygenases in fungal meroterpenoid biosynthetic pathways that use preaustinoid A1 as a common substrate to catalyze divergent rearrangement reactions to form the spiro-lactone in austinol and cycloheptadiene moiety in paraherquonin, respectively. Herein, we report the comparative structural study of AusE and PrhA, which led to the identification of three key active site residues that control their reactivity. Structure-guided mutagenesis of these residues results in successful interconversion of AusE and PrhA functions as well as generation of the PrhA double and triple mutants with expanded catalytic repertoire. Manipulation of the multifunctional Fe(II)/αKG oxygenases thus provides an excellent platform for the future development of biocatalysts.

[1] Graduate School of Pharmaceutical Sciences, The University of Tokyo 7-3-1 Hongo, Bunkyo-ku, Tokyo, 113-0033, Japan. [2] Structural Biology Research Center, Institute of Materials Structure Science, High Energy Accelerator Research Organization KEK, 1-1 Oho, Tsukuba, Ibaraki 305-0801, Japan. [3] Department of Materials Structure Science, School of High Energy Accelerator Science, The Graduate University of Advanced Studies (Soken–dai) 1–1 Oho, Tsukuba, Ibaraki 305-0801, Japan. Correspondence and requests for materials should be addressed to T.S. (email: toshiya.senda@kek.jp) or to I.A. (email: abei@mol.f.u-tokyo.ac.jp)

Meroterpenoids are medicinally important terpenoid-derived hybrid natural products with unusually diverse and complex structures that display a wide range of remarkable biological activities[1–3]. Recent studies on the fungal meroterpenoid biosynthesis identified several key enzymes responsible for the construction of such unique chemical structures[2–4]. Non-heme iron and α-ketoglutarate (αKG; also called 2-oxoglutarate)-dependent dioxygenases are widely distributed in nature, and play a major role in diversifying the molecular scaffold. This superfamily of enzymes employs αKG as a co-substrate and Fe(II) as a cofactor to couple substrate oxidation to concomitant decarboxylation of αKG to form succinate and $CO_2$. The resulting reactive Fe(IV)-oxo species then activates selective C–H bond on the substrate[5–11]. Many of the Fe(II)/αKG-dependent oxygenases found in fungal meroterpenoid biosynthesis are multifunctional and a single enzyme catalyzes a variety of chemistry ranging from simple hydroxylations[12, 13] to remarkable skeletal rearrangements[13–17].

Out of the multifunctional Fe(II)/αKG oxygenases, AusE[14, 18] from austinol (1) biosynthesis in *Aspergillus nidulans* and PrhA[16] from paraherquonin (2) biosynthesis in *Penicillium brasilianum* present a unique case (Fig. 1). AusE ($M_r$ 35,100 protein, 314 amino acids) and PrhA ($M_r$ 35,121 protein, 314 amino acids) are very similar (~78% amino acid sequence identity) αKG oxygenases that both accept preaustinoid A1 (5) as a substrate to form divergent products through dynamic skeletal rearrangement. AusE first desaturates at C1–C2 to form preaustinoid A2 (6), followed by rearrangement leading to the formation of the spirolactone in preaustinoid A3 (7) (Fig. 1, and Supplementary Figs. 1a, b). In contrast, PrhA first desaturates at C5–C6 to form berkeleyone B (11), followed by rearrangement of the A/B-ring to form the cycloheptadiene moiety in berkeleydione (12) (Fig. 1 and Supplementary Fig. 1c). Such functional difference between AusE and PrhA is not apparent from simple comparison of their primary sequences. Thus, comparative structural study of AusE and PrhA highlights important differences that lead to divergent reactions using a common precursor.

While the reactions catalyzed by the multifunctional Fe(II)/αKG oxygenases from fungal meroterpenoid biosynthetic pathways have been characterized, their mechanisms remained unexplored. Here we report the first X-ray crystal structures of AusE and PrhA that provide insight into the multifunctional nature of these enzymes. Comparison of highly homologous enzymes with distinct catalytic functions is a useful approach to decipher their structure–function relationship for rational engineering of enzymes. Through inspection of the structures of these two highly homologous enzymes, we identify important

differences in the active site residues that govern their functions. Structural study and engineering of Fe(II)/αKG oxygenases that catalyze complex multistep oxidations serve as an important milestone in future efforts to understand and control the reactivity of these enzymes.

## Results

**Overall structure of AusE and PrhA**. To investigate the structure–function relationship of AusE and PrhA, we obtained the crystal structures of these two enzymes. Successful crystallization of AusE and PrhA required multiple optimization steps. First, crystallization of AusE required EDTA treatment to remove any contaminating metals that co-purified with the recombinant protein and subsequent anaerobic reconstitution with Mn(II). We succeeded in solving the X-ray crystal structures of AusE complexed with Mn(II) and αKG (AusE-Mn/αKG) at 2.1–2.8 Å resolution (Supplementary Table 1). In vitro analysis showed that EDTA-treated AusE is inactive, and its activity was restored upon addition of Fe(II) but not Mn(II), indicating that AusE does not utilize Mn(II) as a cofactor (Supplementary Fig. 2). Despite numerous attempts, AusE could not be crystallized in the apo-form or in complex with Fe(II).

Next, crystallization of PrhA required replacement of the C-terminal region (Glu295-Val301) with the corresponding AusE residues (Ile295-Ser296-Ser297-Ala298). This C-terminal region plays an important role in intermonomer interactions and PrhA failed to crystallize without this replacement (Supplementary Fig. 3). EDTA-treated PrhA crystallized well under anaerobic conditions, and we succeeded in solving four X-ray crystal structures of PrhA at 2.1–2.3 Å resolution in the apo-form, in complex with Fe(II) (PrhA-Fe), in complex with Fe(II) and αKG (PrhA-Fe/αKG), and in complex with Fe(II), αKG, and the substrate preaustinoid A1 (5) (PrhA-Fe/αKG/5) (Supplementary Table 2 and Supplementary Table 3). Here the metal cofactor, αKG and 5 were introduced into the apo crystal by soaking method.

Both AusE and PrhA exist as symmetric homodimers, which were the same results as a dynamic light-scattering analysis, with a typical double-stranded β-helix fold of non-heme iron oxygenases[19]. The structures of AusE and PrhA resemble those of other fungal FeII/αKG oxygenases such as FtmOx1 (PDB ID: 4ZON, RMSD of 1.6 and 1.2 Å for the 194 $C_\alpha$ and 185 $C_\alpha$ atoms, respectively)[20], and AsqJ (PDB ID: 5DAX, RMSD of 1.1 and 1.0 Å for the 198 $C_\alpha$ and 186 $C_\alpha$ atoms)[21], despite their low primary sequence identity (29–30%). The asymmetric unit of AusE-Mn, AusE-Mn/αKG, and all forms of the PrhA crystals contain two,

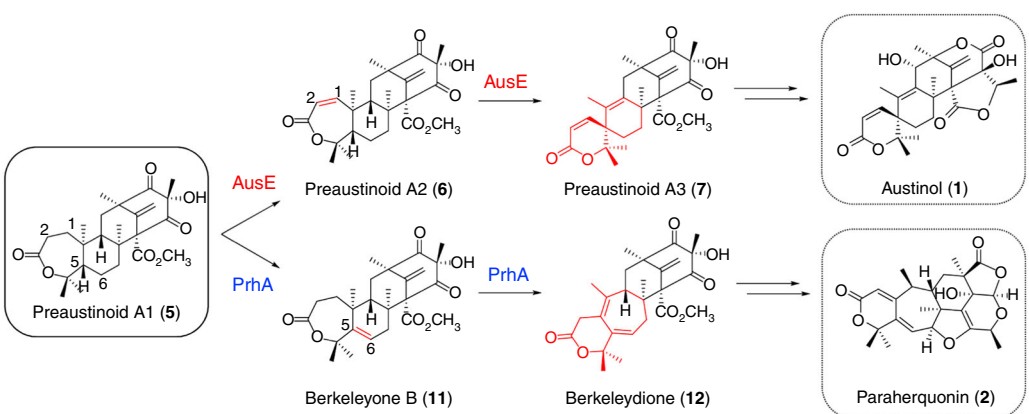

**Fig. 1** Reactions catalyzed by AusE and PrhA oxygenases from preaustinoid A1 (5). AusE and PrhA use preaustinoid A1 (5) as a common substrate to catalyze divergent multistep oxidation reactions in austinol (1) and paraherquonin (2) biosynthetic pathways

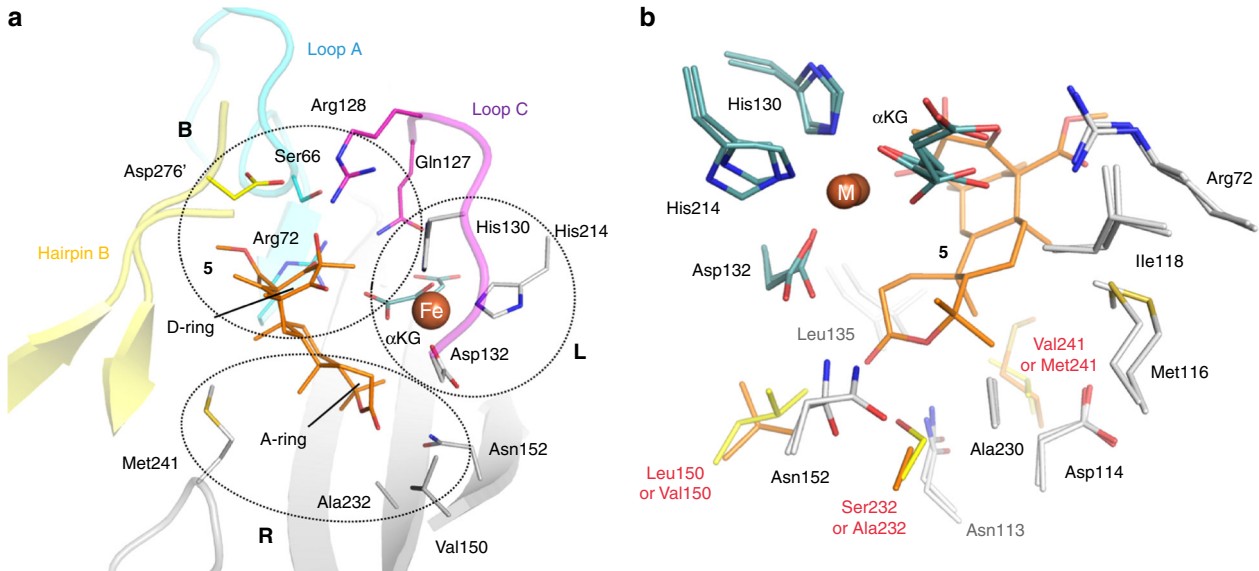

**Fig. 2** The active sites of AusE-Mn/αKG and PrhA-Fe/αKG/**5**. **a** The active site of PrhA-Fe/αKG/**5** showing the residues binding the Fe(II) ion and substrate **5**. The active site is separated into three regions. Region L includes the 2-His-1-Asp triad binding Fe(II). Region B includes the residues interacting with the D-ring of **5**. Finally, Region R includes residues surrounding the A/B-ring of **5**. **b** Comparison of AusE-Mn/αKG and PrhA-Fe/αKG/5 active sites. Selected amino acid residues, co-substrate α-ketoglutarate (αKG, blue) and the substrate **5** (orange) are shown as stick models. Mn(II) or Fe(II) are shown as spheres with the label M. The active site structures of AusE and PrhA are highly similar, except for the three residues (150, 232, and 241) near the A/B-ring of **5**. Residues derived from AusE (Leu150, Ser232, and Val241) and PrhA (Val150, Ala232, and Met241) are colored in yellow and orange, respectively

four, and two molecules (Supplementary Fig. 4). These enzymes have a tight dimer interface that extends ~1500 Å$^2$ (ArealMol program from the CCP4 package)[22] (Supplementary Fig. 5). The enzyme forms a funnel-like reaction chamber conserved in the jelly-roll barrel[23]. As expected, the monomer structures of AusE and PrhA are highly similar with RMSD of 0.6 Å over 248 C$_\alpha$ atoms.

**Active site architectures of AusE and PrhA**. The active sites of AusE and PrhA are strikingly similar with the conserved 2-His-1-Asp facial triad required for metal binding in the αKG oxygenase family (Supplementary Figs. 6 and 7)[5, 19] In the structures of AusE-Mn and PrhA-Fe, His130-Asp132-His214 residues bind the metal ion (Region L in Fig. 2a). In the complex structures (PrhA-Fe/αKG and PrhA-Fe/αKG/**5**), αKG, the facial triad, and a water molecule form an octahedral structure with αKG chelating the iron in a bidentate manner and the water molecule bound *trans* to His130 (Supplementary Fig. 7b). In the active site, αKG interacts with Arg72, Gln127, Thr167, and Arg226 through salt bridge and hydrogen bonding network (Supplementary Fig. 8).

Although we observed iron and αKG in both of PrhA-Fe/αKG/**5**, the substrate **5** was only present in the active site of Chain B (Supplementary Fig. 9a). Consequently, all of the subsequent analyses were performed on Chain B. As predicted, **5** is bound in an orientation that allows oxidation by the Fe(IV)=O species. The structure of **5** in the active site of PrhA matches the previously reported structure of freestanding **5**[24], except for the A-ring conformation. The A-ring of **5** switches from a chair to a boat conformation upon binding to the PrhA active site, and this conformational change might play an important role in determining the initial site of hydrogen atom abstraction by the Fe(IV)-oxo species (Supplementary Fig. 9a). In the complex structures (PrhA-Fe/αKG/**5**), the substrate preaustinoid A1 (**5**) is located across from the His130-Asp132-His214 triad with respect to the Fe(II) ion, and is held in place by hydrogen bonding network (Fig. 2b and Supplementary Fig. 8b). The D-ring of **5**

interacts with the side chain of Arg72 on the β-sheet and Gln127 and Arg128 on Loop C (Region B in Fig. 2a).

Additionally, loop 57–69 (Loop A) and C-terminal hairpin 267′ −280′ of Chain A (Hairpin B) also participate in substrate binding (Region B in Fig. 2a). Asp276′ residue on Hairpin B has a hydrogen bonding interaction with the D-ring of **5** (Supplementary Fig. 10a). Superposition of Chains A and B of the complex structures (PrhA-Fe/αKG/**5**) revealed substantial conformational change for Loop A, which serves as a lid. Upon binding of **5**, Loop A of Chain B moves toward the substrate to switch from the open to the closed state (Supplementary Fig. 10b, c). In the closed state, Cα of Ser66 on Loop A moves 5.4 Å toward the active site to interact with Asp276′ and the D-ring of **5** (Supplementary Fig. 10c). This hydrogen bonding network fixes the flexible Hairpin B in place to allow visualization of the otherwise disordered region. This switch from open to closed state is predicted to be important for protecting the reactive intermediates from the bulk solvent.

Interestingly, the D-ring of **5** is also in direct contact (3.1 Å) with αKG (Supplementary Fig. 11). The binding of αKG to the active site is a prerequisite to the binding of **5**, which implies that this αKG-**5** contact is required for substrate binding. Since αKG is consumed during each reaction cycle, the oxidized product might also leave the active site together with succinate and re-enter upon binding of a new αKG molecule.

**Functional conversion by site-directed mutagenesis**. The fact that AusE and PrhA use identical substrate implies the difference in reactions they catalyze are inherent to the structures of the enzymes. Specifically, we focused on the residues in Region R surrounding the A/B-rings of **5** (Fig. 2a). Careful examination of AusE and PrhA structures showed that almost all of the active site residues in Region R are conserved, except for residues 150, 232, and 241 (Fig. 2b). To test whether this difference is responsible for the divergent reactivity, the two residues that appeared to interact with the A-ring of **5** in PrhA (Val150 and Ala232) were

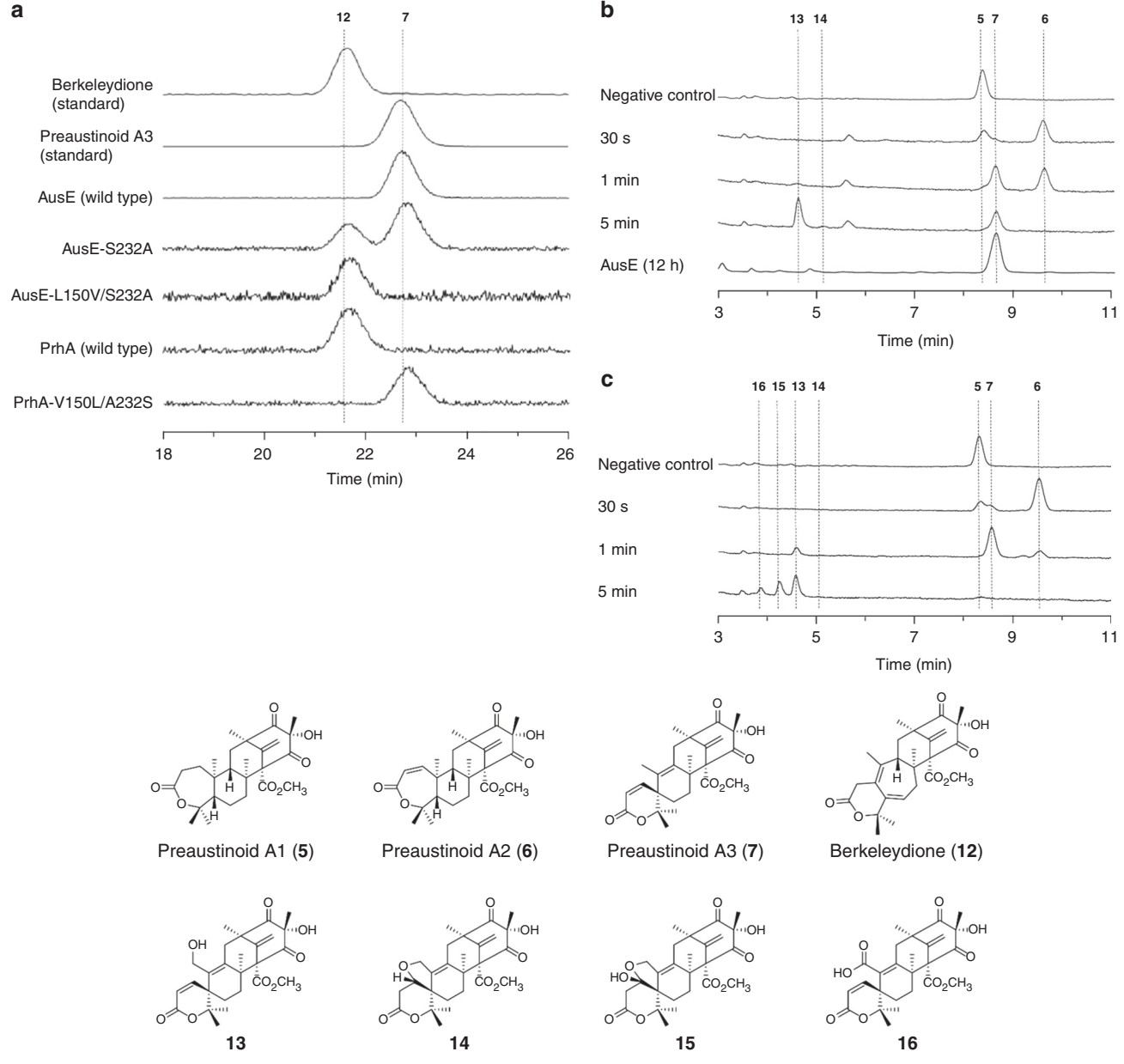

**Fig. 3** In vitro enzyme reactions of the wild-type and mutant AusE and PrhA with preaustinoid A1 (**5**). **a** LC-MS extracted ion chromatograms of **12** and **7** (*m/z* 457.2) in the in vitro enzymatic reactions of **5** and wild-type AusE, AusE-L150V, AusE-L150V/S232A, wild-type PrhA, or PrhA-V150L/A232S. All of the reactions were performed for 5 min. **b**, **c** HPLC chromatograms of in vitro enzymatic reactions of **5** and PrhA-V150L/A232S (**b**) or PrhA-V150L/A232S/V241M (**c**), respectively. The reactions of PrhA-V150L/A232S and PrhA-V150L/A232S/M241V were performed for 30 s, 1 and 5 min. The reaction of wild-type AusE was performed for 12 h as a reference (**b**). New peaks corresponding to **13**–**16** were only observed in the reaction of PrhA-V150L/A232S and PrhA-V150L/A232S/M241V, and not in the reaction of wild-type AusE. HPLC condition optimized for separation of **13**–**16** was used for analysis. **5** incubated in pH 7.5 PIPES buffer for 12 h in the absence of enzyme was used as a negative control. Chromatograms were monitored at 223 nm

replaced with respective residues in AusE (Leu150 and Ser232) and vice versa.

First, single and double mutants of AusE and PrhA were generated through site-directed mutagenesis and their functions were evaluated in vitro using **5** as a substrate. Intriguingly, even a single mutant AusE-S232A gained a PrhA-type "B-ring expansion" activity to produce **12**, while retaining the "spiro-lactone forming" activity to produce **7** (Fig. 3a). Installation of the second mutation generated AusE-L150V/S232A mutant that no longer produced **7**, but yielded the B-ring-expanded **12** as a single product. Similarly, PrhA-V150L/A232S mutant completely lost its original activity, and instead catalyzed the AusE-type "spiro-lactone formation" to yield **7** as a single product. The steady-state

enzyme kinetics analysis using **5** revealed that the two double mutants, PrhA-V150L/A232S and AusE-L150V/S232A, have similar $k_{cat}$ and $K_M$ values to those of the wild-type enzymes (Table 1). This result demonstrates that we succeeded in completely altering the function of the two oxygenases while maintaining the catalytic efficiency of the native enzymes.

Interestingly, prolonged incubation of **5** with PrhA-V150L/A232S yielded two new products (**13** and **14**) (Figs. 3b and 4). The MS analysis indicated that both **13** and **14** possess *m/z* 473 [M + H]⁺, a mass 16 Da larger than that of **7**, supporting that **13** and **14** are oxidized products of **7**. To determine the structures of the newly formed products, **13** and **14** were isolated from large-scale in vitro assays. Isolated **13** spontaneously converted into **14**

### Table 1 The steady-state enzyme kinetics values of the wild-type and mutants of AusE and PrhA to preaustinoid A1 (5)

| Enzyme | $k_{cat}$ (s$^{-1}$) | $K_M$ (μM) | $k_{cat}/K_M$ (s$^{-1}$ μM$^{-1}$) | Product |
|---|---|---|---|---|
| Wild-type AusE | | | | |
| | 19.4 ± 1.8 | 40.9 ± 8.8 | 0.47 | **6** |
| Wild-type PrhA | | | | |
| | 15.4 ± 2.4 | 48.3 ± 16.1 | 0.32 | **11** |
| PrhA-V150L/A232S | | | | |
| | 15.8 ± 2.5 | 43.0 ± 15.2 | 0.25 | **6** |
| AusE-L150V/S232A | | | | |
| | 20.4 ± 2.1 | 37.0 ± 10.2 | 0.55 | **11** |

These values were calculated by analyzing the consumption of substrate **5** by HPLC analysis, as described in the "Methods" section

in the absence of enzyme, and its structure was deduced to be a C-13 hydroxylated derivative of **7**, based on the structure of **14** with an additional tetrahydrofuran ring (Fig. 4).

Encouraged by the production of further oxidized compounds by the PrhA-V150L/A232S mutant, a third M241V mutation was installed to generate the PrhA triple mutant. PrhA-V150L/A232S/M241V mutant also catalyzes the same rearrangement reaction as AusE and generates **13** as well as two additional products **15** and **16** (Fig. 3c and Supplementary Fig. 12). Based on MS and NMR analysis, **15** was identified as C-1 hydroxylated derivative of **14** (Fig. 4). On the other hand, structural analysis of derivatized **16** indicated that **16** is a C-13 carboxylated analog of **7** (Fig. 4). Thus, the PrhA double mutant is capable of catalyzing additional oxidation step, and the triple mutant further oxidizes **13** to **16** and **14** to **15** compared to the wild-type AusE. **13–16** are unnatural meroterpenoid derivatives that have not been isolated previously. The detailed structural determination of **13–16** by HRMS and NMR are described in Supplementary Note 1.

In addition to **5**, the multifunctional AusE also accepts berkeleyone A (**3**)[25] as a substrate to catalyze sequential oxidations to produce preaustinoid A (**4**), 5-hydroxyberkeleyone A (**8**), preaustinoid C (**9**), and austinoid C (**10**) as shunt products (Fig. 4 and Supplementary Fig. 1d). We performed in vitro assay with the PrhA wild-type and mutants using **3** as a substrate (Supplementary Fig. 13). While the wild-type PrhA did not accept **3** as a substrate, PrhA-V150L/A232S mutant yielded **8** as a major product and **10** as a minor product through hydroxylation at C-5. In contrast, the activity PrhA-V150L/A232S/M241V mutant was almost identical to that of the wild-type AusE (Supplementary Fig. 13).

**Crystal structures of the PrhA mutants**. To understand the structural basis for the expanded catalytic functions of the PrhA mutants, we also solved the X-ray crystal structures of PrhA-V150L/A232S mutant in complex with Fe(II), αKG, and its substrate (**3**, **5**, **6**, or **7**) at 2.25–2.3 Å resolution (Fig. 5c–e and Supplementary Fig. 14), as well as PrhA-V150L/A232S/M241V mutant in complex with Fe(II), αKG, and **3** at 2.35 Å resolution (Fig. 5f and Supplementary Tables 3 and 4). In the crystal structures, the overall structures of the enzymes, the positions of Fe(II) ion, αKG, as well as the octahedral metal coordination geometry, are almost identical to those of the wild-type PrhA. Further, the binding positions of the four substrates do not change significantly and the D-ring region of the substrates is fixed by the same hydrogen bonding network (Fig. 5a). In contrast, the A/B-ring region has little interaction with the enzyme active site, which explains how the enzyme accommodates substrates with variable A/B-ring structures to catalyze multistep oxidation. The D-ring tethered to the flexible Loop A and Hairpin

B is mobile as indicated by the relatively high B-factor values. On the other hand, the movement of the A/B-ring embedded deeper inside the hydrophobic active site is limited to permit hydrogen atom abstraction at only one or two carbon centers for each substrate.

The oxidation reactions catalyzed by the non-heme iron oxygenases are initiated by selective hydrogen atom abstraction step. Theoretically, the active site undergoes change from six- to five-coordinate state upon substrate binding, and the activated molecular oxygen bind to the ferrous ion *trans* to any of the three ligands of the 2-His-1-Asp triad[26]. While we cannot predict the exact location of oxygen binding, substrate proximity to the Fe center provides key information in understanding the enzyme mechanism based on structural data. In the case of PrhA, hydrogen atom abstraction at C-5 leads to desaturation of the B-ring to form **11** (Supplementary Fig. 1c). On the other hand, AusE and PrhA-V150L/A232S mutant first abstracts C-2 hydrogen atom to desaturate the A-ring to form **6** (Supplementary Fig. 1a). Comparison of the structures of the PrhA wild-type and PrhA-V150L/A232S mutant in complex with **5** showed that C-2 moved closer toward the iron center in the double mutant (4.2 vs. 4.4 Å) (Fig. 5b, c). Here A232S mutation pushes C-3 and C-2 toward the iron center through a hydrogen bond with the C-3 carbonyl of **5** to facilitate the abstraction of H-2 (Fig. 5g). In the absence of Ser232, the A-ring of **5** is more likely to undergo boat-to-chair flip, which brings C-5 closer toward the iron center to facilitate H-5 abstraction by the wild-type PrhA.

Next, AusE and PrhA-V150L/A232S mutant catalyzes oxidation of **6** to form the spirocyclic **7** through radical rearrangement of the A/B-ring. Here the C-5 atom of **6** (5.2 Å) is closer to the Fe(II) center compared to the C-9 atom (6.0 Å) (Fig. 3d), which supports the mechanism involving initial abstraction of H-5 by the Fe(IV)-oxo species (Supplementary Fig. 1b). Subsequently, the spirocycle formation can proceed through two possible paths. Path A involves radical rebound to form the hydroxylated intermediate, which leads to deprotonation at C-9 and migration of C-1 to eliminate water to form **7**. Path B involves homoallyl–homoallyl radical rearrangement through the cyclo-propylcarbinyl radical intermediate[16]. The lack of potential catalytic base near H-9 and the stereoelectronic problem regarding the *syn*-arrangement of the migrating bond and the leaving group suggest that Path B involving homoallyl–homoallyl radical rearrangement to be more plausible.

PrhA-V150L/A232S catalyzes further oxidation of C-13 to form additional product **13**. Complex structure of PrhA-V150L/A232S and preaustinoid A3 (**7**) showed that C-10/C-13 bond of the six-membered spiro-lactone **7** is rotated ~30° clockwise compared to the seven-membered lactone **5** to orient C-13 closer toward the Fe(II) ion at a distance of 7.4 Å (Fig. 5e). In the reaction of the PrhA mutants and **3**, hydrogen atom abstraction occurs competitively at C-3 or C-5 to yield **4** or **8**, respectively (Fig. 4). In the complex structures of PrhA-V150L/A232S and PrhA-V150L/A232S/M241V with berkeleyone A (**3**), both H-3 and H-5 are well oriented to react with the reactive ferry intermediate (Fig. 5f and Supplementary Fig. 14).

## Discussion

Structure–function analyses of the two multifunctional Fe(II)/αKG oxygenases, AusE from *A. nidulans* and PrhA from *P. brasilianum*, revealed that subtle differences in the active site structure result in dramatic change in the reaction outcome. Most importantly, we identified three key amino acid residues (150, 232, and 241) lining the substrate-binding pocket to achieve the functional interconversion of AusE and PrhA. By exchanging the corresponding residues, AusE-L150V/S232A and PrhA-V150L/

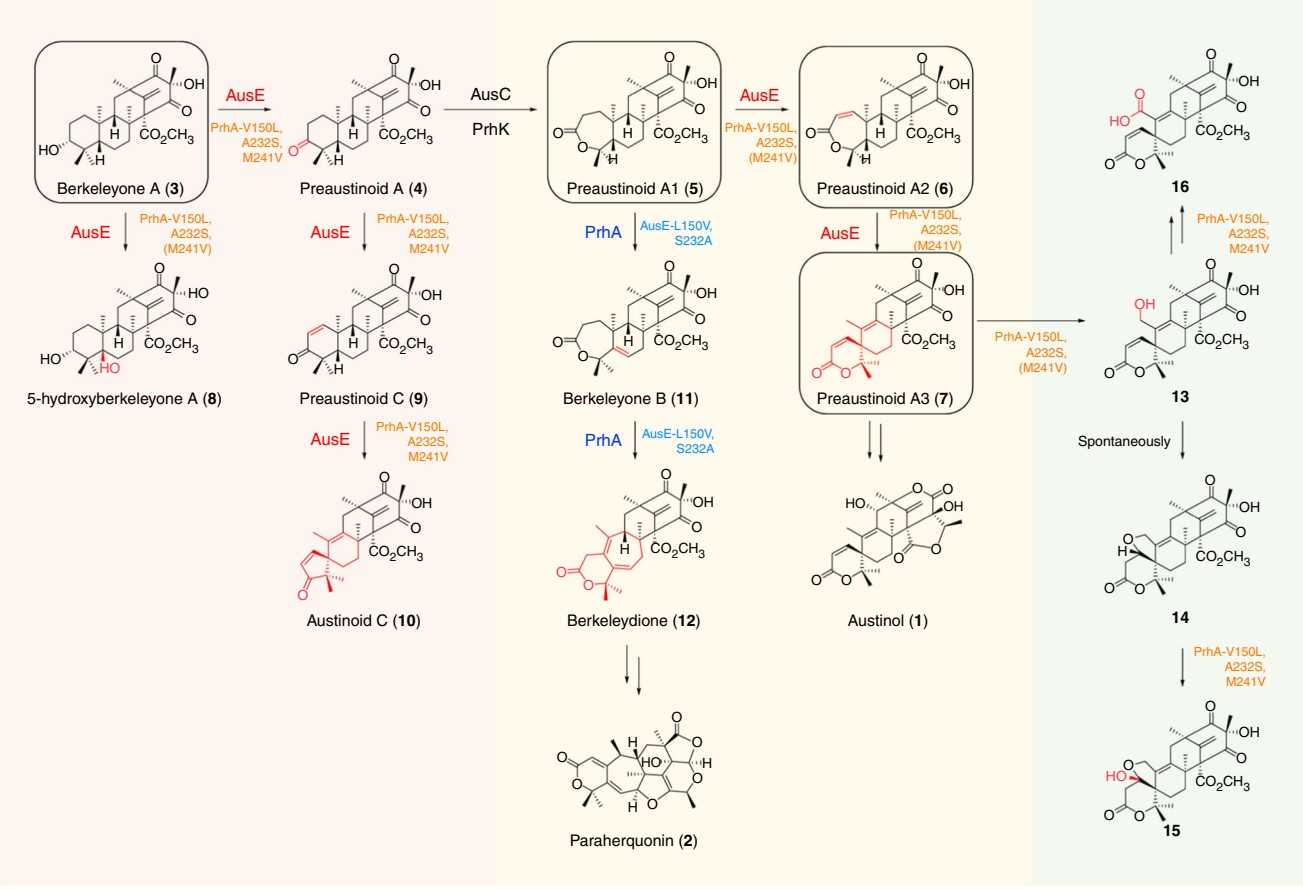

**Fig. 4** Reactions catalyzed by multifunctional AusE and PrhA oxygenases. Left block (red) summarizes the reactions catalyzed by AusE or PrhA-V150L/A232S/(M241V) from berkeleyone A (**3**). Middle block (yellow) summarizes the reactions catalyzed by AusE, PrhA, AusE-L150V/S232A, and PrhA-V150L/A232S/(M241V) from preaustinoid A1 (**5**). Right block (green) summarizes the reactions catalyzed by PrhA-V150L/A232S/(M241V) from preaustinoid A3 (**7**) to generate unnatural compounds (**13–16**). Substrates used to obtain complex structures are circled

A232S double mutants lost their original functions and gained the ability to yield the B-ring-expanded berkeleydione (**12**) and spiro-lactone preaustinoid A3 (**7**), respectively, from preaustinoid A1 (**5**) (Fig. 4). To our knowledge, this is the first example of successful functional conversion of αKG oxygenases that catalyze dynamic skeletal rearrangement reactions.

Based on the in vitro analysis, Ser232 plays an important role in determining the initial site of hydrogen atom abstraction at either C-2 or C-5 of **5**. One possible function of Ser232 is to fix the A-ring conformation of the substrate in place to prevent chair-boat flip to promote hydrogen abstraction at C-2. When the A-ring of **5** is in the chair conformation, C-5 is predicted to move closer to the iron center to facilitate hydrogen abstraction at this site. As indicated by the kinetics data, the catalytic efficiencies of the double mutants are comparable to those of the wild-type enzymes, further corroborating the success of functional conversion. It is intriguing that only two or three active site residues modulate the multifunctional activities of the two enzymes. Similar fine-tuning of a bifunctional αKG oxygenase has been achieved through few active site mutations for deacetoxycephalosporin/deacetylcephalosporin C synthase (DAOC/DACS) in β-lactam biosynthesis[27]. In both this work and the DAOC/DACS work, comparison of highly conserved enzymes illuminated the residues that influence the catalytic functions.

Interestingly, we have recently found that AusE′ in *Penicillium brasilianum* NBRC 6234 with 92% sequence identity to PrhA catalyzes AusE-type spiro-lactone formation despite its higher primary sequence homology to PrhA (Supplementary Fig. 15)[16].

Phylogenetic tree analysis also indicated AusE′ is closer to PrhA than AusE (Supplementary Fig. 16). Inspection of AusE′ sequence revealed that the three key active site residues of this enzyme, Ile150, Ser232, and Val241, resemble those of AusE (Supplementary Fig. 15), which further support our current finding that these three residues are important for controlling the enzyme activity.

The PrhA double and triple mutants, to our surprise, acquired expanded catalytic repertoires to catalyze iterative oxidation reactions to afford a series of unnatural molecules (Fig. 4). The complex structure of PrhA-V150L/A232S double mutant and **7** indicated that unlike other substrates, the C-13 of **7** points toward the iron center to allow hydrogen abstraction. Moreover, the M241V mutation is thought to provide additional space around C-13 to allow further oxidation to generate **13–16** (Supplementary Fig. 17). The primary function of the enzymes is to generate radical species through activation of selective substrate C–H bond. The subsequent skeletal rearrangement might proceed spontaneously upon radical generation depending on the structure and conformation of the substrate without elaborate enzymatic assistance. Nonetheless, it is quite remarkable that a single enzyme catalyzes such multistep and dynamic chemistry.

Another important observation includes the presence of Loop A and Hairpin B that serve as lids upon substrate binding. The loop and hairpin regions are highly disordered in the absence of substrate. Indeed, there is substantial conformational movement (~112° rotation around $C_\alpha$–$C_\beta$ of Ser66) in Loop A to form hydrogen bonds with the substrate (Supplementary Fig. 10). The

"open" form of the enzyme thus allows the substrate to enter the active site, while the "closed" form binds the substrate via hydrogen bonds including those between Ser66 and Asp276' and encloses the radical species during catalysis.

In addition to the crystal structures of the wild-type enzymes, we acquired the structures of the PrhA-V150L/A232S double mutant in complex with four substrates (**3**, **5**, **6**, or **7**) and the PrhA-V150L/A232S/M241V triple mutant in complex with **3** to

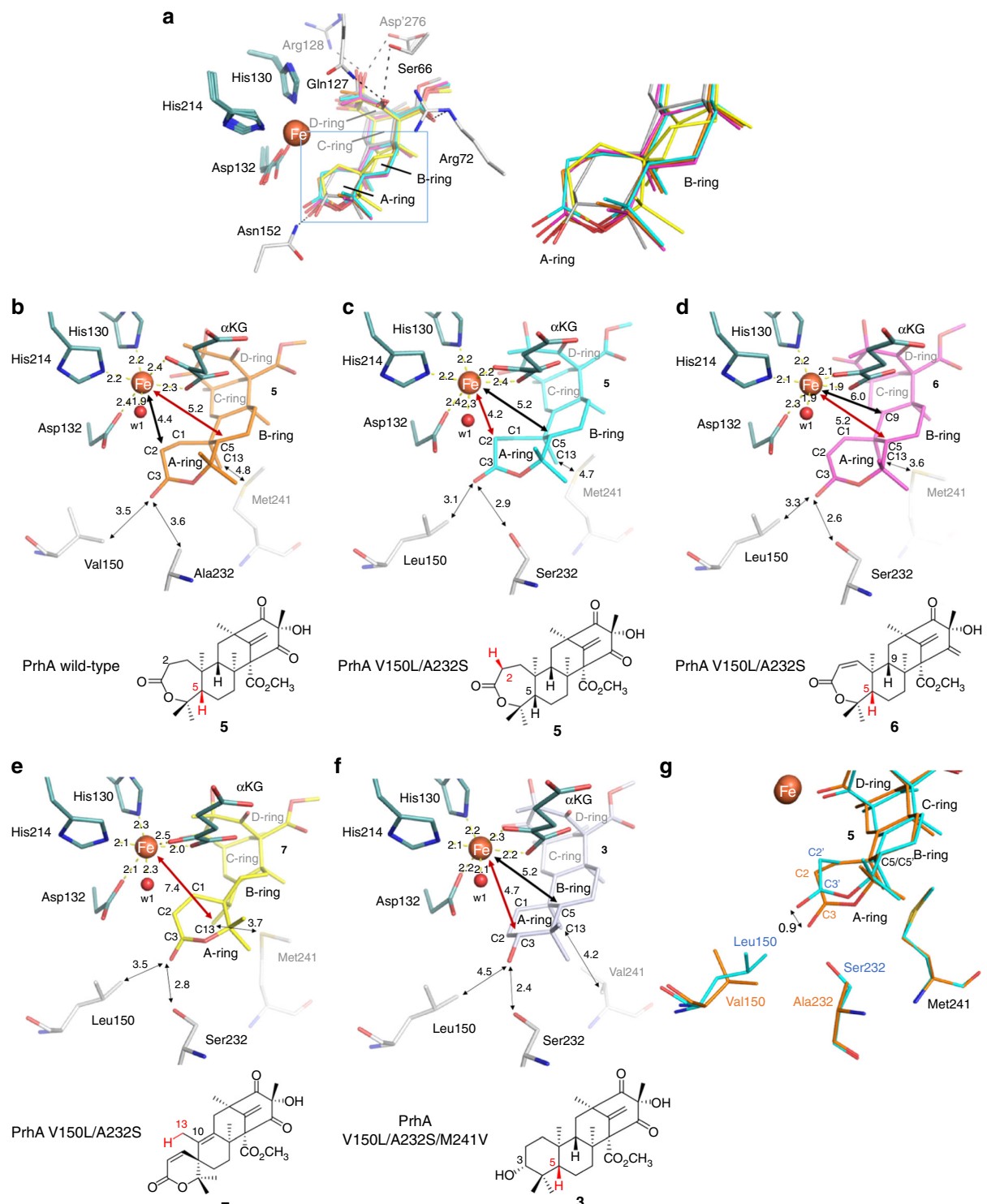

**Fig. 5** Comparison of the active site structures of the wild-type and the mutant PrhA. **a** Superimposition of the active site of PrhA-Fe/αKG/**5**, PrhA(V150L/A232S)-Fe/αKG/**5**, PrhA(V150L/A232S)-Fe/αKG/**6**, PrhA(V150L/A232S)-Fe/αKG/**7**, and PrhA(V150L/A232S/M241V)-Fe/αKG/**3**. Substrates in each structure are colored in orange, cyan, magenta, yellow, and white, respectively. Close-up view of the boxed A/B-ring region is shown on the right. Amino acid residues interacting with **5** in PrhA-Fe/αKG/**5** structure are shown as stick model (white). **b**–**f** Close-up views of the active site of PrhA-Fe/αKG/**5** (**b**), PrhA(V150L/A232S)-Fe/αKG/**5** (**c**), PrhA(V150L/A232S)-Fe/αKG/**6** (**d**), PrhA(V150L/A232S)-Fe/αKG/**7** (**e**), and PrhA(V150L/A232S/M241V)-Fe/αKG/**3** (**f**). **g** Superimposition of the active site of PrhA-Fe/αKG/**5** and PrhA(V150L/A232S)-Fe/αKG/**5**

follow the course of multistep oxidation. The substrates in the complex structures are highly similar, especially in the C/D-ring region of the substrate, which is firmly fixed by hydrogen bonding network (Fig. 3a). In contrast, the A/B-ring region is surrounded by hydrophobic residues and only forms hydrogen bond with Asn152 (Fig. 5a). This loose substrate recognition in the A/B-ring region allows substrates with variable A/B-ring structures to bind and yield a series of unnatural molecules by our engineered enzyme. Additionally, the movement of Loop A and Hairpin B might affect the positioning of the A/B-ring to the active Fe=O species to facilitate hydrogen abstraction at various sites.

The structural basis for these remarkable multifunctional non-heme iron oxygenases is rather limited, except for the seminal crystallographic studies on isopenicillin N synthase and related enzymes by Baldwin, Schofield, and co-workers, and a recent report on AsqJ by Groll and co-workers[21, 28–31]. Here we successfully solved the complex structures of multifunctional Fe(II)/αKG oxygenases in fungal meroterpenoid biosynthetic pathways, and our structure-guided enzyme engineering provided solid basis for how these multifunctional enzymes catalyze iterative and sequential oxidation reactions by accommodating substrates with highly diverse A/B-ring scaffolds.

Generally, enzymes with wide substrate scope are ideal targets for engineering of biocatalysts through directed evolution. Enzymes in the secondary metabolic pathways tend to be relatively promiscuous compared to those in primary metabolism. From evolutionary standpoint, cost-effective generation of chemical diversity in secondary metabolism is advantageous for the survival of producing organisms to screen molecules with desired bioactivity[32]. In the case of the multifunctional oxygenases in fungal meroterpenoid pathways, a single highly versatile αKG oxygenase rapidly diversifies the structures of complex molecules. Manipulation of the catalytic repertoire of the multifunctional αKG oxygenases thus provides an excellent platform for the future development of biocatalysts and production of medicinally important molecules for drug discovery.

## Methods

**Materials**. Oligonucleotide primers were chemically synthesized by Eurofins genomics. Other chemicals and analytical grade solvents were purchased from Wako Chemicals Ltd. (Tokyo, Japan). HPLC was performed on Shimadzu Prominence HPLC system using TSK-gel ODS-80Ts column (Tosoh Co. Ltd., 4.6 i. d. × 150 mm) or COSMOSIL 5C$_{18}$-MS-II column (Nacalai Tesque Inc. Ltd., 10 i. d. × 250 mm). LC-MS analysis was performed on a Shimadzu Prominence HPLC system equipped with Bruker Compact qTOF mass spectrometer fitted with an electrospray ionization source using a COSMOSIL πNAP column (Nacalai Tesque, 2.0 i.d. × 150 mm). NMR spectra were recorded on a Bruker AVANCE III HD 600 MHz ($^1$H, 600 MHz; $^{13}$C, 150 MHz), a Bruker Avance III HD 900 MHz ($^1$H, 900 MHz; $^{13}$C, 225 MHz), and a JEOL ECX-500 ($^1$H, 500 MHz; $^{13}$C, 125 MHz) NMR spectrometers.

**Preparation of AusE and PrhA substrates**. Berkeleyone A (**3**), preaustinoid A2 (**6**), and preaustinoid A3 (**7**) were isolated from the culture of the *Aspergillus oryzae* NSAR1 ($niaD^-$, $sC^-$, $\Delta argB$, and $adeA^-$) transformants expressing AusB, AusE, and AusC. *A. oryzae* NSAR1 transformants co-expressing PrhI and PrhJ was used to prepare preaustinoid A1 (**5**). The culture medium was extracted with ethyl acetate, concentrated, and purified by silica-gel chromatography (Wakogel C-200, 0 to 5% methanol in chloroform). The fraction containing the target compound was concentrated and purified by Shimadzu Prominence HPLC system using TSK-gel ODS-80TM column (Tosoh Co. Ltd., 7.8 i.d. × 300 mm, 50% acetonitrile, 3.0 ml min$^{-1}$)[16, 18].

**Enzyme expression and purification for crystallization**. The DNA encoding N-terminal truncated AusE-(6P-A298) was amplified by PCR using pET28a_AusE-(M1-A298) plasmid as a template and ausE-6–298-f and ausE-6–298-r primers containing NdeI and BamHI recognition sites, respectively (Supplementary Table 6). The DNA encoding N-terminal truncated and C-terminal chimeric PrhA-(6P-A298) was amplified by PCR using pET22b_PrhA-(M1-V301) plasmid as a template and prhA-6-298-f and prhA-6-298-r primers containing NdeI and BamHI recognition sites, respectively (Supplementary Table 6). DNA fragments were digested with NdeI and BamHI (Takara) and ligated into pET28a vector

(Novagen). *Escherichia coli* Rosetta$^{TM}$ 2(DE3)pLysS (Novagen) was selected as an expression host.

Rosetta$^{TM}$ 2(DE3)pLysS was employed as an expression host. *E. coli* Rosetta cells harboring pET28a_AusE-(M1-A298) or pET22b_PrhA-(M1-V301) plasmids were cultivated in Luria–Bertani medium supplemented with chloramphenicol (12.5 μg ml$^{-1}$) and kanamycin (25 μg ml$^{-1}$) at 37 °C with shaking. At OD$_{600}$ = 0.6, 0.3 mM Isopropyl β-D-1-thiogalactopyranoside was added to the cultures induce the gene expression and incubated for at 20 °C for 16 h.

Cells from the cultures were pelleted at 5,000 × $g$ for 10 min at 4 °C, resuspended in 50 mM Tris-HCl (pH 7.5), 200 mM NaCl, 5% (v/v) glycerol, and 5 mM imidazole buffer (buffer L), and lysed by sonication. After centrifugation at 12,000 × $g$ for 30 min, the supernatant was applied to a pre-equilibrated HisTrap HP column (4 °C, 5 ml, GE Healthcare). The loaded column was washed with 50-column volume of buffer L containing 10 mM imidazole, and the proteins were eluted with buffer L containing 10–300 mM imidazole. The pooled fraction containing the desired protein was then applied to a HiTrap Q HP column (4 °C, 5 ml, GE Healthcare), and eluted with a 0–1.0 M NaCl gradient in 50 mM Tris-HCl (pH 8.0). To the pooled fractions containing the desired protein, 1 mM ethylenediaminetetraacetic acid was added and the solution was rotated at 4 °C for 1 h. Then, the protein solution was concentrated and applied to HiLoad 16/60 Superdex 200 prepacked gel filtration column (4 °C, GE Healthcare), eluted with 20 mM Tris-HCl (pH 7.5), 100 mM NaCl, and 2 mM dithiothreitol buffer, and concentrated to 15 mg ml$^{-1}$ using Amicon Ultra-4 (MWCO: 10 kDa, Millipore) at 4 °C. The protein concentration was calculated by measuring UV absorption at 280 nm and using the extinction coefficients of AusE (996 M$^{-1}$ cm$^{-1}$) and PrhA (0.754 M$^{-1}$ cm$^{-1}$).

Dynamic light-scattering analysis of the purified wild-type AusE and PrhA displayed a monomodal distribution with polydispersity values of 18.2 and 21.1% and predicted molecular masses of 72 and 75 kDa, respectively. Since the calculated values of both enzymes are 35.1 kDa, the recombinant AusE and PrhA exist as homodimers.

**Crystallization and structure determination**. The following procedures were all performed under anaerobic conditions[33]. Well-diffracting AusE crystals were obtained at 20 °C, in 20% (w/v) PEG4000, 100 mM sodium citrate (pH 5.6), 10 mM MnCl$_2$, and 15 mg ml$^{-1}$ of purified AusE with a vapor diffusion method. PrhA, PrhA-V150L/A232S, and PrhA-V150L/A232S/M241V crystals were obtained at 20 °C, in 27% (w/v) PEG3350, 200 mM lithium citrate, and 15 mg ml$^{-1}$ of purified enzymes with a vapor diffusion method. PrhA crystals were soaked in 25% (w/v) PEG3350 and 200 mM lithium citrate solution (solution A), containing 10 mM FeSO$_4$, 50 mM αKG, and 5 mM **5** for 6 h at 20 °C. PrhA-V150L/A232S crystals were soaked in solution A with 10 mM FeSO$_4$, 50 mM αKG, and 10 mM **5** for 10 h at 20 °C. PrhA-V150L/A232S crystals were soaked in solution A with 10 mM FeSO$_4$, 50 mM αKG, and 5 mM **3**, **6**, or **7** for 6 h at 20 °C. PrhA-V150L/A232S/M241V crystals were soaked in solution A with 10 mM FeSO$_4$, 50 mM αKG, and 5 mM **3** for 6 h at 20 °C. The crystals were moved into the reservoir solution or the soaking solution with 25% (v/v) glycerol for 10 s for cryoprotection and flash-cooled in liquid nitrogen. BL-1A and BL-17A (Photon Factory, Tsukuba, Japan) were used for obtaining X-ray diffraction data sets. We used wavelengths of 1.8926 Å (Mn absorption edge)[34] for single-wavelength anomalous diffraction (SAD) of wild-type AusE. The other data were collected under the wavelengths of 0.9800 or 1.1000 Å.

XDS[35] and AIMLESS[36] were used for data processing and scaling. The Mn substructure determination and the phase calculation were performed by the SAD method with AutoSol in PHENIX[37, 38]. Initial model of AusE was constructed by AutoBuild in PHENIX[38, 39]. On the other hand, the initial phases of the PrhA, PrhA-V150L/A232S, and PrhA-V150L/A232S/M241V structures (Phaser in PHENIX)[40] determined by molecular replacement using AusE structure—a search model. Model building was performed by Coot[41] and refined with Phenix.Refine[42].

The three-dimensional models of **3**, **5**, **6**, and **7** were calculated with the Chem3D Ultra 13 (CambridgeSoft). The occupancy of the substrate was maintained at 1.0 during refinements, because the B-factor values of ligand atoms were similar to the surrounding amino acid residues. The final crystal data and intensity statistics are listed in Supplementary Tables 2, 3, and 4. The cavity volumes were predicted using CASTP server (http://cast.engr.uic.edu/cast/). PyMOL (http://www.pymol.org) was used to prepare all structure representations.

**HPLC and LC-MS analysis of in vitro assays**. The in vitro assay samples were analyzed by HPLC using a TSK-gel ODS-80T$_S$ column (Tosoh Co. Ltd., 4.6 i.d. × 150 mm, 1.0 ml min$^{-1}$, 50% CH$_3$CN/H$_2$O with 0.5% acetic acid). The absorbance was monitored at 210 nm. LC-MS analysis was performed using a COSMOSIL® πNAP column (Nacalai Tesque, 2.0 i.d. × 150 mm, 0.30 ml min$^{-1}$, 35% CH$_3$CN/ H$_2$O with 0.05% formic acid).

**Metal dependency analysis of EDTA-treated AusE activity**. The metal dependency of AusE enzyme reaction was performed using berkeleyone A (**3**) as a substrate. Each reaction was performed in a total volume of 50 μl containing 50 mM Tris-HCl buffer (pH 7.5), 100 μM of **3**, 4 mM ascorbate, 5 mM αKG, and 10 μM AusE, with or without 10 mM FeSO$_4$ or MnCl$_2$ at 30 °C for 3 h. Each reaction

was quenched by addition of 50 μl of methanol and vortexed. After centrifugation, the supernatant was analyzed by HPLC (Supplementary Fig. 2). Non-EDTA-treated AusE was prepared following the enzyme purification method described above, but without the addition of ethylenediaminetetraacetic acid after the HiTrap Q HP column purification step.

**Enzymatic reaction assay for LC-MS analysis**. The enzymatic reactions of AusE, AusE-S232A, and AusE-L150V/S232A with preaustinoid A1 (**5**) were performed in a total volume of 50 μl containing 10 μM enzyme, 100 μM **5**, 50 mM Tris-HCl buffer (pH 7.5), 4 mM ascorbate, 0.2 mM $FeSO_4$, and 5 mM αKG for 5 min at 30 °C. The enzymatic reactions of PrhA and PrhA-V150L/A232S with **5** were performed in a total volume of 50 μl containing 10 μM enzyme, 100 μM **5**, 50 mM PIPES buffer (pH 7.5), 4 mM ascorbate, 0.2 mM $FeSO_4$, and 5 mM αKG at 30 °C for 5 min. The reaction mixtures were analyzed by LC-MS (Fig. 3a).

**Enzymatic reaction assay for HPLC analysis**. All enzymatic reactions were performed in a total volume of 50 μl containing 10 μM enzyme, 100 μM substrate, 50 mM PIPES buffer (pH 7.5), 4 mM ascorbate, 0.2 mM $FeSO_4$, and 5 mM αKG at 30 °C. Each reaction was quenched by addition of 50 μl of methanol and vortexed. After centrifugation, the supernatant was analyzed by HPLC. The enzymatic reactions of PrhA-V150L/A232S or PrhA-V150L/A232S/M241V with preaustinoid A1 (**5**) were incubated for 30 s, 1 and 5 min (Fig. 3b, c). The enzymatic reaction of AusE with **5** was performed incubated 12 h (Fig. 3b). The enzymatic reactions of PrhA-V150L/A232S/M241V with compound **14**, **15**, or mixture of **13** and **14** were also performed for 12 h (Supplementary Fig. 12). The enzymatic reactions of PrhA, PrhA-V150L/A232S, PrhA-V150L/A232S/M241V, or AusE with berkeleyone A (**3**) were performed for 1 h (Supplementary Fig. 13).

**Steady-state enzyme kinetics**. **5** (8, 20, 50, 100, and 250 μM) and 0.1 μM purified enzyme were used to determine the kinetic parameters of AusE, AusE-L150V/S232A, PrhA, or PrhA-V150L/A232S with **5**. Each assay was performed in a total volume of 50 μl containing 50 mM Tris-HCl buffer (pH 7.5), 4 mM ascorbate, 0.2 mM $FeSO_4$, and 5 mM αKG at 30 °C for 1 min, and quenched by the addition of 50 μl methanol. Consumption of substrate **5** was analyzed with HPLC by calculating the total peak areas of substrate. Each reaction was done in triplicate. GraphPad Prism (GraphPad Prism Software Inc., San Diego, CA) was used for statistics data analysis (Table 1).

**Site-directed mutagenesis**. Site-directed mutagenesis of AusE (S232A and L150V/S232A) and PrhA (V150L/A232S and V150L/A232S/M241V) were performed using a QuikChange Site-Directed Mutagenesis Kit (Stratagene). Following primer pairs were used for installing each mutation; L150V (ausE-L150V-f and ausE-L150V-r), S232A (ausE-S232A-f and ausE-S232A-r), V150L (prhA-V150L-f and prhA-V150L-r), A232S (prhA-A232S-f and prhA-A232S-r), and M241V (prhA-M241V-f and prhA-M241V-r) (Supplementary Table 6). The pET28a plasmid with *ausE*-(M1-A298) insert and the pET22b plasmid with *prhA*-(M1-V301) insert were used as templates for the mutagenesis reactions. The resulting plasmids were transformed into *E. coli* Rosetta™ 2(DE3)pLysS. All mutants were expressed and purified following the method described in section "Expression and purification of AusE and PrhA".

**Data availability**. The data supporting the findings of this study are available within the article and its Supplementary Information Files or from the corresponding authors on reasonable request. Protein Data Bank (PDB): The coordinates and the structure factor amplitudes for the apo structure of PrhA, AusE, PrhA, and its mutants complexed with metals and ligands were deposited under accession codes 5YBL, 5YBM, 5YBN, 5YBO, 5YBP, 5YBQ, 5YBR, 5YBS, and 5YBT.

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

## Acknowledgements

We thank Dr Yudai Matsuda for critical reading of the manuscript. This work was supported in part by a Grant-in-Aid for Scientific Research from the Ministry of Education, Culture, Sports, Science and Technology, Japan (JSPS KAKENHI Grant Number JP15H01836 and JP16H06443), and JSPS Research Fellowships for Young Scientists (to Y.N.), the Platform for Drug Discovery, Informatics, and Structural Life Science (PDIS), and Basis for Supporting Innovative Drug Discovery and Life Science Research (BINDS) from Japan Agency for Medical Research and Development (AMED). The synchrotron radiation experiments were performed at the BL-1A and BL-17A of the Photon Factory with proposal Nos. 2014T006 and 2015G530, respectively.

## Author contributions

Y.N., T.S., and I.A. designed the experiments. Y.N., T.M., T.A., and S.H., performed the experiments. Y.N., T.M., H.N., M.S., T.S., and I.A. analyzed the data. Y.N., H.N., T.A., T.S., and I.A. wrote the paper.

## Additional information

**Competing interests:** The authors declare no competing financial interests.

