## [Peer Review File · Nature Communications]

REVIEWERS' COMMENTS:

Reviewer #1 (Remarks to the Author):

This MS reports new and very detailed information both about the mechanisms and about the rational engineering of non-heme iron enzymes active in secondary metabolism. This class of enzymes is not as well understood as, for example, cytochrome P450 oxygenases, but are widespread and involved in fascinating and useful oxidative transformations including complex rearrangements in plants, bacteria fungi and marine organisms. The work has been performed to a very high standard and the MS is presented also at a very high level. The work concerns two parallel NHI enzymes involved in the oxidative processing of fungal meroterpenoids. The expression, purification and crystal structure determination is very difficult and the high quality results in this area are rarely achieved, and thus the methods and results reported will be of interest to many involved in structural determination as well as biosynthetic investigations. The mutants generated shed significant light on the mechanisms of the oxidations and how the enzyme controls the balance between all the different possible modes of oxygenation and rearrangement. The experiments are clearly designed and the results are generally clearly shown in the MS and supporting data. The MS tells a strong and coherent story and I found no significant scientific problems in the data interpretation or conclusions. Some minor points could be addressed.

1. The introduction suggests that these NHI enzymes are mostly involved in fungal meroterpenoid biosynthesis - actually they are very widely distributed and NHI rearrangements are relatively common.

2. I found the lack of detailed chemical mechanisms (e.g. traditional arrow pushing!) in the actual MS disappointing. The textual descriptions of the mechanisms are good, but support from clear diagrams for both the WT and mutant reactions (e.g as shown partially in Fig S1) would improve the impact and readability of the MS for non-expert readers. In particular discussion of stereoselectivity (i.e the faces over which the iron-oxo moiety has access) would be pertinent.

Reviewer #2 (Remarks to the Author):

The authors compare AusE and PrhA, two non-heme oxygenases that catalyze distinct reactions in terpenoid biosynthesis. The authors successfully crystallized the two enzymes and performed mutagenesis that swapped activities of the two enzymes, and also generated enzyme mutants with new activity. The manuscript is well-written and easy to follow. All required information for protein structure is present. This work is an important demonstration of how these iron containing enzymes can be manipulated and is suitable for Nature Comms.

I have the following minor comments:

page 5: "Despite numerous attempts, AusE could not be crystallized in the apo-form or in complex with Fe(II)."

Did the authors try anaerobic conditions for this? I am not suggesting that the authors attempt this in a revision, but it would be informative to know if all crystallization attempts were performed in air.

page 14: reference 22 is to a software program, which I think is a mistake

Figure 3: Can the authors add structures to this figure, so that it is not required to go to one of the other figures to match the compound number to the structure?

I think that it is essential to include a phylogenetic tree of fungal non-heme iron oxygenases that shows the placement of AusE and PrhA in a larger context. I would expect that these two enzymes would be closely related.

Reviewer #3 (Remarks to the Author):

Overall I thought this was a well written and interesting manuscript describing biochemical, structural and engineering studies on an unusual oxygenase structured in terpenoid biosynthesis. The manuscript is generally well written (needs a final proofing). Although, there is a previous structure for one of the oxygenases, the current work is much more substantial and provides new insights. Overall, I'm in favour of publication with a few minor modifications, mostly to the figures.

Some of the Chemdraw type figures could be larger e.g. p28.

Supp fig 1, or an edited version thereof could be in main text.

Label N-/E termini on figures 3,4,5

Important to add in solution data on oligomerization status (I may have missed this), i.e. using gel-filtration or SEC MALS.

Discuss role of the C-terminus in the text.

Specific points:

P3 para 2 – 'highly identical' to 'very similar'

Perhaps mention structure function studies on bacterial α -KG oxygenases in the introduction, e.g. work has been done on the bifunctional enzymes of cephalosporin biosynthesis (though I note mentioned in Discussion)

P5 – To investigate....

What is the oligomerization state of the protein in solution – gel filtration/SEC-MACS data should be provided.

Is the C-termini involved in dimerisation/catalysis?

What is the occupancy of the substrate?

Show electron density for the water/metal in Supp Fig 7

In Supp Fig 6, there appears to be density close to the α -KG carboxylate? What is this – show density for metal + water (use stereoview figures if need be).

P6 – what exactly is meant by suprafacial? (and free standing).

Please label chair/boat as appropriate in Supp Fig 9

P7 – 'direct contact' – give distance

P8 – Define the assays being used (and in legend to Table 1) (it is important these analyse product formation as described in methods).

P10 – I don't think its correct to imply the initial H-abstraction controls the reaction – it's not always rate limiting and the products can vary even with the same initial abstraction.

Response to Reviewers' comments:

REVIEWERS' COMMENTS:

Reviewer #1 (Remarks to the Author):

This MS reports new and very detailed information both about the mechanisms and about the rational engineering of non-heme iron enzymes active in secondary metabolism. This class of enzymes is not as well understood as, for example, cytochrome P450 oxygenases, but are widespread and involved in fascinating and useful oxidative transformations including complex rearrangements in plants, bacteria fungi and marine organisms. The work has been performed to a very high standard and the MS is presented also at a very high level. The work concerns two parallel NHI enzymes involved in the oxidative processing of fungal meroterpenoids. The expression, purification and crystal structure determination is very difficult and the high quality results in this area are rarely achieved, and thus the methods and results reported will be of interest to many involved in structural determination as well as biosynthetic investigations. The mutants generated shed significant light on the mechanisms of the oxidations and how the enzyme controls the balance between all the different possible modes of oxygenation and rearrangement. The experiments are clearly designed and the results are generally clearly shown in the MS and supporting data. The MS tells a strong and coherent story and I found no significant scientific problems in the data interpretation or conclusions. Some minor points could be addressed.

1. The introduction suggests that these NHI enzymes are mostly involved in fungal meroterpenoid biosynthesis - actually they are very widely distributed and NHI rearrangements are relatively common.

Response: Thank you for the instructive comment. According to the suggestion, we have changed the sentence to “**Non-heme iron and α -ketoglutarate (α KG; also called 2-oxoglutarate) dependent dioxygenases are widely distributed in nature, and play a major role in diversifying the molecular scaffold.**” (page 3, line 6 from the top).

2. I found the lack of detailed chemical mechanisms (e.g. traditional arrow pushing!) in the actual MS disappointing. The textual descriptions of the mechanisms are good, but support from clear diagrams for both the WT and mutant reactions (e.g as shown partially in Fig S1) would improve the impact and readability of the MS for non-expert readers. In particular discussion of stereoselectivity (i.e the faces over which the iron-oxo moiety has access) would be pertinent.

Response: According to the suggestion, we substantially revised Supplementary Figure 1.

1) Fixed arrow pushing to appropriate one. 2) Described stereochemistry to the hydrogen atom abstracted by the iron-oxo moiety. 3) Added mutant names which catalyze the same reactions as corresponded WT to the subheadings of Fig S1 a, b, c, and d respectively. 4) Added detail catalytic reactions for constructing compound 13, 15, and 16, as Supplementary Figure 1e, f, and g respectively.

Reviewer #2 (Remarks to the Author):

The authors compare AusE and PrhA, two non-heme oxygenases that catalyze distinct reactions in terpenoid biosynthesis. The authors successfully crystallized the two enzymes and performed mutagenesis that swapped activities of the two enzymes, and also generated enzyme mutants with new activity. The manuscript is well-written and easy to follow. All required information for protein structure is present. This work is an important demonstration of how these iron containing enzymes can be manipulated and is suitable for Nature Comms.

I have the following minor comments:

page 5: "Despite numerous attempts, AusE could not be crystallized in the apo-form or in complex with Fe(II)." Did the authors try anaerobic conditions for this? I am not suggesting that the authors attempt this in a revision, but it would be informative to know if all crystallization attempts were performed in air.

Response: We are sorry for the confusion, but in the original text, we had already stated that "All of the following procedures were performed under anaerobic conditions." (page 19, line 3 from the top).

page 14: reference 22 is to a software program, which I think is a mistake

Response: Thank you. We have changed into the appropriate reference "21" according to the suggestion (page 14, line 24 from the top).

Figure 3: Can the authors add structures to this figure, so that it is not required to go to one of the other figures to match the compound number to the structure?

Response: We have added structures to Figure 3.

I think that it is essential to include a phylogenetic tree of fungal non-heme iron oxygenases that shows the placement of AusE and PrhA in a larger context. I would expect that these two enzymes would be closely related.

Response: According to the suggestion, we have newly added Supplementary Figure 16 of a phylogenetic tree.

Reviewer #3 (Remarks to the Author):

Overall I thought this was a well written and interesting manuscript describing biochemical, structural and engineering studies on an unusual oxygenase structured in terpenoid biosynthesis. The manuscript

is generally well written (needs a final proofing). Although, there is a previous structure for one of the oxygenases, the current work is much more substantial and provides new insights. Overall, I'm in favour of publication with a few minor modifications, mostly to the figures.

Some of the Chemdraw type figures could be larger e.g. p28.

Response: According to the suggestion, we have improved the sizes of chemdraw type figures larger.

Supp fig 1, or an edited version thereof could be in main text.

Response: We also agree. However, because of the limited space, we cannot make it. Instead, according to the suggestion of reviewer #1 as mentioned above, we substantially revised Supplementary Figure 1.

Label N-/E termini on figures 3,4,5

Response: Yes, we have added label N/C termini on Supplementary Figure 3, 4, and 5.

Important to add in solution data on oligomerization status (I may have missed this), i.e. using gel-filtration or SEC MALs.

Response: According to the suggestion, we have changed the sentence to “Both AusE and PrhA exist as symmetric homodimers, which were the same results as a dynamic light-scattering analysis, with a typical double-stranded β -helix fold of non-heme iron oxygenases.” (page 5, line 22 from the top) and added the sentence “A dynamic light-scattering analysis after the gel filtration of wild-type AusE and PrhA revealed a monomodal distribution, with a polydispersity value of 18.2% and 21.1%, and an estimated molecular mass of 72 kDa and 75 kDa respectively. On the other hand, the calculated value of the both enzymes 35.1 kDa. These observations indicated that the recombinant AusE and PrhA are a homodimeric enzymes.” (page 18, line 20 from the top).

Discuss role of the C-terminus in the text.

Response: We are sorry for the confusion, we had already discussed Hairpin B region on C-termini is related to the binding of substrates. “Another important observation includes the presence of Loop A and Hairpin B that serve as lids upon substrate binding. The loop and hairpin regions are highly disordered in the absence of substrate. Indeed, there is substantial conformational movement ($\sim 112^\circ$ rotation around $C\alpha$ - $C\beta$ of Ser66) in Loop A to form hydrogen bonds with the substrate (Supplementary Fig. 10). The “open” form of the enzyme thus allows

the substrate to enter the active site, while the “closed” form binds the substrate via hydrogen bonds including those between Ser66 and Asp276’ and encloses the radical species during catalysis.” (page 14, line 3 from the top).

Specific points:

P3 para 2 – ‘highly identical’ to ‘very similar’

Response: We have changed the sentence to “**very similar**” (page 3, line 17 from the top) according to your suggestion.

Perhaps mention structure function studies on bacterial α -KG oxygenases in the introduction, e.g. work has been done on the bifunctional enzymes of cephalosporin biosynthesis (though I note mentioned in Discussion)

Response: We would like to keep the introduction as is, since we mention it in the Discussion. “**The structural basis for these remarkable multifunctional non-heme iron oxygenases is rather limited, except for the seminal crystallographic studies on isopenicillin N synthase and related enzymes by Baldwin, Schofield, and co-workers, and a recent report on AsqJ by Groll and co-workers.**” (page 14, line 21 from the top).

P5 – To investigate....

Response: According to the suggestion, we have changed the sentence to “**To investigate**” (page 5, line 3 from the top).

What is the oligomerization state of the protein in solution – gel filtration/SEC-MACS data should be provided.

Response: As mentioned above, we have changed the sentence to “**Both AusE and PrhA exist as symmetric homodimers, which were the same results as a dynamic light-scattering analysis, with a typical double-stranded β -helix fold of non-heme iron oxygenases.**” (page 5, line 22 from the top) and added the sentence “**A dynamic light-scattering analysis after the gel filtration of wild-type AusE and PrhA revealed a monomodal distribution, with a polydispersity value of 18.2% and 21.1%, and an estimated molecular mass of 72 kDa and 75 kDa respectively. On the other hand, the calculated value of the both enzymes 35.1 kDa. These observations indicated that the recombinant AusE and PrhA are a homodimeric enzymes.**” (page 18, line 20 from the top) according to your suggestion.

Is the C-termini involved in dimerisation/catalysis?

Response: We have discussed Hairpin B region on C-termini is related to the binding of substrates. Please refer to the following sentences, “Another important observation includes the presence of Loop A and Hairpin B that serve as lids upon substrate binding. The loop and hairpin regions are highly disordered in the absence of substrate. Indeed, there is substantial conformational movement (~112° rotation around C α -C β of Ser66) in Loop A to form hydrogen bonds with the substrate (Supplementary Fig. 10). The “open” form of the enzyme thus allows the substrate to enter the active site, while the “closed” form binds the substrate via hydrogen bonds including those between Ser66 and Asp276’ and encloses the radical species during catalysis.” (page 14, line 3 from the top).

What is the occupancy of the substrate?

Response: According to the suggestion, we have added the sentence “The occupancy of the substrate was maintained at 1.0 during refinements, because the B-factor values of ligand atoms were close to around amino acid residues.” (page 20, line 9 from the top).

Show electron density for the water/metal in Supp Fig 7

In Supp Fig 6, there appears to be density close to the α -KG carboxylate? What is this – show density for metal + water (use stereoview figures if need be).

Response: According to the suggestion, we have shown the electron density of water/metal according to your suggestion in Supplementary Figure 6 and Supplementary Figure 7 respectively. Some density around α KG looks like background which appeared during calculation of the omit map.

P6 – what exactly is meant by suprafacial? (and free standing).

Response: We have deleted the word “suprafacial”.

Please label chair/boat as appropriate in Supp Fig 9

Response: We have added the labels of chair/boat to the A ring in Supplementary Figure 9.

P7 – ‘direct contact’ – give distance

Response: Done. “Interestingly, the D ring of 5 is also in direct contact (3.1 Å) with α KG (Supplementary Figure 11).” (page 7, line 20 from the top).

P8 – Define the assays being used (and in legend to Table 1) (it is important these analyse product

formation as described in methods).

Response: We have subdivided the items of the method in order to get easy to define the assays being used (From page 20, line 16 to page 23, line 11). We have added the sentence “**These values were calculated by analyzing the consumption of substrate 5 by HPLC analysis, as described in the Methods.**” in legend to Table 1.

P10 – I don’t think its correct to imply the initial H-abstraction controls the reaction – it’s not always rate limiting and the products can vary even with the same initial abstraction.

Response: We appreciate the thoughtful comment by the reviewer. We deleted “control”, and have changed the sentence to “**The oxidation reactions catalyzed by the non-heme iron oxygenases are initiated by selective hydrogen atom abstraction step.**” (page 10, line 18 from the top).